# A Brief Overview of the Epigenetic Regulatory Mechanisms in Plants

**DOI:** 10.3390/ijms26104700

**Published:** 2025-05-14

**Authors:** Theodoros Tresas, Ioannis Isaioglou, Andreas Roussis, Kosmas Haralampidis

**Affiliations:** 1Section of Botany, Biology Department, National and Kapodistrian University of Athens, 15772 Athens, Greece; dtresas@biol.uoa.gr (T.T.); aroussis@biol.uoa.gr (A.R.); 2Bioscience Program, Biological and Environmental Science and Engineering Division, King Abdullah University of Science and Technology (KAUST), Thuwal 23955, Saudi Arabia; ioannis.isaioglou@kaust.edu.sa

**Keywords:** abiotic stress, biotic stress, seed development, seed germination, DNA methylation, histone modifications, ubiquitin ligase complex activities, miRNA, lncRNA

## Abstract

Plants continuously adapt to their environments by responding to various intrinsic and extrinsic signals. They face numerous biotic and abiotic stresses such as extreme temperatures, drought, or pathogens, requiring complex regulatory mechanisms to control gene activity and adapt their proteome for survival. Epigenetic regulation plays a crucial role in these adaptations, potentially leading to both heritable and non-heritable changes across generations. This process enables plants to adjust their gene expression profiles and acclimate effectively. It is also vital for plant development and productivity, affecting growth, yield, and seed quality, and enabling plants to “remember” environmental stimuli and adapt accordingly. Key epigenetic mechanisms that play significant roles include DNA methylation, histone modification, and ubiquitin ligase complex activity. These processes, which have been extensively studied in the last two decades, have led to a better understanding of the underlying mechanisms and expanded the potential for improving agriculturally and economically important plant traits. DNA methylation is a fundamental process that regulates gene expression by altering chromatin structure. The addition of methyl groups to cytosines by DNA methylases leads to gene suppression, whereas DNA demethylases reverse this effect. Histone modifications, on the other hand, collectively referred to as the “histone code”, influence chromatin structure and gene activity by promoting either gene transcription or gene silencing. These modifications are either recognized, added, or removed by a variety of enzymes that act practically as an environmental memory, having a significant impact on plant development and the responses of plants to environmental stimuli. Finally, ubiquitin ligase complexes, which tag specific histones or regulatory proteins with ubiquitin, are also crucial in plant epigenetic regulation. These complexes are involved in protein degradation and play important roles in regulating various cellular activities. The intricate interplay between DNA methylation, histone modifications, and ubiquitin ligases adds complexity to our understanding of epigenetic regulation. These mechanisms collectively control gene expression, generating a complex and branching network of interdependent regulatory pathways. A deeper understanding of this complex network that helps plants adapt to environmental changes and stressful conditions will provide valuable insights into the regulatory mechanisms involved. This knowledge could pave the way for new biotechnological approaches and plant breeding strategies aimed at enhancing crop resilience, productivity, and sustainable agriculture.

## 1. Introduction

The concept of epigenesis was first introduced by Aristotle and later expanded upon by William Harvey in 1650, who coined the term “epigenetics” and defined it as a “gradual development process that increases complexity from initially homogeneous material found in different animals’ eggs” [1]. In 1942, Waddington redefined epigenetics as “the whole complex of developmental processes” between the “genotype and phenotype” [2]. With the advent of molecular genetics and the identification of molecular mechanisms able to redesign gene expression, the contemporary notion of epigenetics sums up to reversible mitotic and meiotic changes in gene expression without alterations in gene content [3,4,5]. Although plants are stationary organisms, their environment constantly changes. This exposes them to various external stimuli and signals, necessitating different responses. As a result, plants need to continuously adapt their genetic potential to cope with these changes. They have evolved sophisticated molecular mechanisms to control gene expression and modify their active proteome, especially when dealing with environmental stresses like interactions with microorganisms (symbiotic and pathogenic) and changes in abiotic factors such as salinity, temperature, sunlight, drought, flooding, cold, and heat. This enables plants to synchronize their development and metabolic processes with these challenges, respond to developmental cues, and cope with stress [6,7].

Plant epigenetic regulation orchestrates heritable or non-heritable changes in gene expression without altering the DNA sequence itself [8]. The adaptation of plants is mainly achieved through the activation or silencing of gene transcription by recruiting several molecular epigenetic processes. These processes include DNA methylation, histone modification through methylation and ubiquitination, and the regulation of gene transcription by non-coding RNAs (Figure 1). Additionally, the interaction among these epigenetic processes is complex, dynamic, and not yet fully understood. For instance, the interplay between DNA methylation and histone modifications is essential for regulating gene expression, maintaining chromatin structure, and determining cellular identity, while the interplay between miRNAs and ubiquitination can influence the stability and activity of proteins associated with miRNAs, especially Argonaute proteins, which are crucial for miRNA function [8,9,10]. Recent advancements in laboratory techniques and high-throughput approaches—such as genomics, transcriptomics, and proteomics—have significantly expanded our understanding of plant epigenetic mechanisms. This has further enhanced developments in plant science at the cellular and molecular levels, ultimately benefiting humanity [11].

DNA methylation is a crucial molecular mechanism that directly affects gene activation and/or silencing, transcription levels, and chromatin structure and condensation [12]. In all eukaryotic organisms, including plants, DNA methylation occurs at cytosine residues of the nucleotide sequences, often in the context of CG, CHG, and CHH, through the action of DNA methyltransferases [9]. These enzymes add methyl groups to cytosines, leading to the suppression of gene expression, associated with transposons, imprinting, heterosis, seed development, and fruit ripening [13]. Conversely, DNA demethylases, like the ROS1 family, remove these methyl groups, allowing for the dynamic regulation of gene expression [14,15].

Transcription activation and reverse gene silencing are further regulated by the “histone code”, which involves various histone modifications such as mono-, di-, or triple methylation of lysine and arginine residues of centromere histones [16]. This epigenetic mechanism involves enzymatic proteins called “readers”, “writers”, and “erasers” that methylate and demethylate histones, regulating gene expression at the chromatin level. Similar histone modification occurs through the addition and removal of acetyl groups, with acetyltransferases and methyltransferases playing crucial roles. Histone acetylation typically results in an open chromatin structure and active gene transcription, while histone methylation can either activate or repress transcription, depending on the modified residues and biological context [17]. Growing evidence emphasizes the significant role of histone modifications in plant development and responses to environmental stimuli. Moreover, histone modifications act as environmental memory, helping plants adapt to changes in their surroundings [18]. In addition to the well-characterized histone modifications, other components of the “histone code” are also being explored, including the relatively novel post-translational modifications (PTMs) of histone succinylation and lactylation. Histone succinylation, which occurs at lysine residues, is generally associated with gene activation and has been documented in animal cells, although reports in plants have thus far focused on non-histone proteins [19,20,21]. Histone lactylation, a more recent discovery, involves the addition of a lactyl group from lactic acid to lysine residues and is also linked to gene activation [22]. While lactate production during fermentation under stress conditions, such as hypoxia, has been observed in plants [23], no studies have yet confirmed histone lactylation in these organisms. However, the evolutionary conservation of histone modification machinery suggests that this modification may also occur in plants under similar stress conditions.

Histone modification also occurs through ubiquitination, which involves adding a ubiquitin molecule to specific lysine residues of histones, primarily H2A and H2B, or creating a chain of ubiquitin molecules (multiubiquitination) [24]. Ubiquitin ligases are crucial for recognizing substrates during the ubiquitination process. Ubiquitination and ubiquitin ligase complexes are increasingly recognized as important factors in plant epigenetic regulation [25,26,27]. E3 ligases function through a complex interactive mechanism involving E1, E2, and E3 coordination, which will be detailed later. Tagging specific histones with ubiquitin molecules can lead to protein degradation via the proteasome pathway and, in coordination with previously described histone modifications, results in functional alterations and nucleosome localization. For instance, the monoubiquitination of histone H2A by the Polycomb Repressive Complex 1 (PRC1) leads to specific gene silencing in plants [28,29]. Other E3 ubiquitin ligase complexes regulate stress responses by influencing the stability and accumulation of stress-responsive transcription factors or downstream signaling cascades [24,30]. These processes play significant roles in regulating various molecular activities in plants, such as hormone signal transduction, flowering time control, and their resistance to external stress [31]. The interplay and coordination of the various forms of histone modifications mentioned above fine-tune gene expression [32]. Furthermore, DNA methylation is often linked to histone H3 lysine 9 dimethylation (H3K9me2), resulting in a repressive chromatin state, while ubiquitin ligases have been shown to interact with various epigenetic regulators and microRNAs. These interactions form complex regulatory networks that can affect the stability and activity of histone-modifying enzymes, thereby influencing chromatin structure and gene transcription [33].

Agriculture and biotechnology face major challenges in addressing food security, climate change, and sustainable practices. Traditional methods often cannot meet the rising global food demand while minimizing environmental impacts. Although genetically modified (GM) crops and CRISPR gene editing have improved crop yields and pest resistance, their adoption is limited by regulatory and societal barriers. Furthermore, traditional breeding is slow and imprecise in developing crops that withstand stresses like drought and pests. This review highlights key epigenetic mechanisms that regulate plant growth, development, and responses to environmental stress. This knowledge offers valuable insights and has significant applications in biotechnology, enabling genetic engineering and synthetic biology approaches to enhance desirable traits in crops, such as increased yield and resilience to environmental challenges. Furthermore, it informs agricultural practices by supporting the development of improved crop varieties and precision agriculture techniques, ultimately contributing to sustainable food production and security in a rapidly changing environment.

## 2. DNA Methylation

DNA methylation involves the addition of a methyl group to cytosine bases in DNA, resulting in the formation of 5-methylcytosine. In plants, cytosine methylation occurs in symmetric CG and CHG contexts, and in the asymmetric CHH context (where H = A, C, or T) [34]. Methylation predominantly occurs in the CG context, followed by CHG and CHH contexts [35]. This methylation pattern has been observed in various plants, including Arabidopsis (CG: 24%, CHG: 6.7%, CHH: 1.7%), cassava (CG: 58.7%, CHG: 39.5%, CHH: 3.5%), soybean (CG: 63%, CHG: 44%, CHH: 5.9%), maize (CG: 65%, CHG: 50%, CHH: 5%), and rice (CG: 54.7%, CHG: 37.3%, CHH: 12%). Studies also indicate that gene body methylation primarily occurs in the CG context [36], whereas transposable elements (TEs), a type of mobile genetic element, exhibit high methylation levels across all three sequence contexts [13]. Plant genome DNA methylation generally favors chromatin stability and transcriptional gene silencing (TGS), while demethylation produces gene activation [37].

In plants, DNA methylation is directed by the canonical RNA-directed DNA methylation (RdDM) pathway. This pathway follows a bifurcated multistep process, involving small interfering RNAs (siRNAs) and non-coding RNAs (ncRNAs), which together form a functional complex capable of inducing DNA methylation [38]. In summary, this complex pathway consists of the following basic steps [34]:Initial biogenesis of siRNA and ncRNA premature transcripts by RNA-polymerase II (POL II).Processing of pre-RNAs via a complicated and compartmentalized pathway involving DICER (DCL3) and ARGONAUTE (AGO4 and AGO6) systems.Recruitment of POL-IV and POL-V, two plant-specific RNA polymerases.Formation of the active ribonucleoprotein AGO-siRNA-ncRNA-POL V complex, which recruits DRM2 (domains rearranged methyltransferase 2), directed for specific methylation marked by the readers SUVH2 and SUVH9.

It should be noted that different homologues produce counter effects, indicative of the fine-tuning mechanisms for appropriate gene regulation. For example, while readers SUVH2 and SUVH9, members of the DNA methyl-reader SU(VAR)3-9 family, participate in the main scheme for gene silencing, readers SUVH1 and SUVH3 are utilized for specific gene activation sites [13,15,35,39]. A variation in the canonical RdDM pathway, termed non-canonical RdDM, involves small RNAs from different origins (including viral RNA), initially spliced and processed by a different route before entering the Argonaut system of maturation. These small RNAs may lead to differential readings of methylation-targeted sites [40,41].

Genome-wide methylation homeostasis is maintained by target-specific demethylases, which function by replacing 5-methylcytosine residues with unmethylated cytosine. In *Arabidopsis*, there are four known DNA demethylase enzymes: Demeter (DME), Repressor of Silencing 1 (ROS1), Demeter-Like 2 (DML2), and DML3. While the precise biological roles of DML2 and DML3 remain unclear, DME is essential for demethylating the maternal allele and regulating gene imprinting in the endosperm, whereas ROS1 is involved in plant developmental regulation and responses to biotic and abiotic stresses. Studies on *ROS1* mutants (either hypomethylated or hypermethylated) in *Arabidopsis* accordingly modify the plant’s resistance to *Hyaloperonospora*. Furthermore, the expression of *ROS1* is influenced by RdDM activity; reduced DNA methylation suppresses ROS1 expression, while increased DNA methylation promotes *ROS1* expression. This regulatory mechanism suggests a potential role of ROS1 in plant resistance responses [14,42,43].

The maintenance of DNA methylation is dependent on the sequence context. In the CG context, as observed in *Arabidopsis*, maintenance is facilitated by METHYLTRANSFERASE 1 (MET1) [44], while in the CHG context, maintenance is carried out by CHROMOMETHYLASE 2 and 3 (CMT2 and CMT3) [45]. For the CHH context, maintenance is ensured by either CMT2 or the RdDM pathway. *Arabidopsis* mutants deficient in the maintenance of methylation in specific contexts exhibit enhanced resistance to pathogens. For instance, mutants lacking methylation maintenance in the CG context (met1 mutant), in both CG and CHH contexts (met1 nrdp2 mutant), and in CHG and CHH contexts (triple ddc mutant of drm1 drm2 cmt3) display improved resistance against *Pseudomonas syringae* [46]. Additionally, the ddc mutant also demonstrates enhanced resistance to the downy mildew-causing pathogen *Hyaloperonospora arabidopsidis* [47]. Table 1 summarizes well-described methylation regions in *Arabidopsis* and their associated functions resulting from this epigenetic effect.

## 3. Histone Modifications

The regulatory epigenetic mechanisms of methylation and demethylation occur not only at the DNA level but also extend to the histone proteins of the nucleosome. The framework that regulates gene expression through specific chemical modifications to histone proteins, the key components of chromatin structure in eukaryotic cells, is known as the “histone code” [49]. This code includes various post-translational modifications (PTMs) of histones, including acetylation, methylation, phosphorylation, and ubiquitination. These modifications determine how accessible DNA is to the transcriptional machinery and thus significantly influence gene expression. Histones are the basic units of chromatin, and different modifications can either loosen or tighten chromatin packaging. For instance, acetylation typically results in a more open chromatin configuration, known as euchromatin, which facilitates gene activation [50]. In contrast, certain patterns of methylation can create a more compact structure, referred to as heterochromatin, which inhibits transcription [51]. The interaction between these modifications not only affects chromatin architecture but also works in conjunction with other epigenetic factors. This dynamic interplay contributes to the regulation of gene expression across various cellular contexts and developmental stages. Therefore, the histone code is essential for understanding how genetic information is expressed and regulated without changing the underlying DNA sequence. Developmental and environmental cues trigger these chromatin modification processes, which directly regulate specific gene expression and silencing [52]. In the following sections, these mechanisms will be examined in detail.

### 3.1. Histone Lysine Methylation

Similar to DNA methylation, which occurs on cytosine residues, histone methylation specifically targets lysine residues on histones. In this case, histone proteins are methylated at specific lysine and/or arginine residues. The methylation process is carried out by “readers”, “writers”, and “erasers”, each representing distinct enzymatic actions (Table 2). The “readers” are proteins that recognize the methylated lysine and arginine residues. “Writers” are specific histone methyltransferases that possess SET domains and are responsible for adding methyl groups to these identified residues. During this methylation process, one, two, or three methyl groups can be added, resulting in mono-, di-, or tri-methylation. The removal of these methyl groups is carried out by two families of demethylases known as “erasers”, the Jumonji C (JmjC) domain-containing proteins and Lysine-Specific Demethylase 1 (LSD1)-like proteins [53]. Additionally, this process is finely regulated by the POLYCOMB REPRESSIVE COMPLEXES 1 and 2 (PRC1 and PRC2), which specifically methylate lysine residue 27 of histone H3 [54]. Typically, the methylation occurs at lysine residues 3, 4, 9, and 27, and at arginine residues 2, 8, 17, and 26 of histone H3. It can also occur at position 3 of histone H4. Each of these modifications affects gene expression and developmental stages in distinct ways. Most research in this area uses the plant model *Arabidopsis thaliana*, which is beneficial due to its small genome, short life cycle, mutation tolerance, and well-understood post-embryonic organogenesis [55].

H3K9 mono- and dimethylations are typically found in heterochromatin regions that contain highly repetitive sequences and in transposons. This methylation contributes to maintaining the condensed, repressive state of heterochromatin. The process involves three partially redundant methyltransferases: KRYPTONITE (KYP)/SUVH4, SUVH5, and SUVH6. In *Arabidopsis*, a plant-specific triple tandem Agenet factor known as Agenet-containing Protein 1 (ADCP1) or Agenet domain-containing Protein 1 (AGDP1) has been identified as the primary reader of H3K9me2. The ADCP1/AGDP1 complex plays a crucial role in modulating the silencing of transposable elements and regulating the level of condensation of heterochromatin [51]. Heterochromatin is also associated with histone H3 lysine 27 monomethylation (H3K27me1), performed by two plant-specific histone methyltransferases, ARABIDOPSIS TRITHORAX-RELATED PROTEIN 5 (ATXR5) and ATXR6 [56].

The loss of H3K27me1 function results in the decondensation of heterochromatin and silencing of transposable elements. On the other hand, the trimethylated form of H3K27 (H3K27me3), which involves the addition of three methyl groups, serves a repressive function but is primarily found in euchromatin regions. H3K27me3 is generally associated with the entire transcribed region of inactive genes. Its deposition is regulated by the evolutionarily conserved Polycomb Repressive Complex 2 (PRC2). This evidence suggests that distinct mechanisms and enzymes are involved in the regulation of heterochromatin and euchromatin, each playing different roles in gene expression by targeting various histone modifications and lysine residues. The role of the PRC2 complex in various stages of plant development was demonstrated in *Arabidopsis*, which has an expanded family of PRC2 subunits that are differentially expressed at different stages of plant growth [13,57,58]. These are as follows:Three homologs of E(z): CURLY LEAF (CLF), SWINGER (SWN), and MEDEA (MEA).Three homologs of Su(z)12: EMBRYONIC FLOWER 2 (EMF2), VERNALIZATION 2 (VRN2), and FERTILIZATION-INDEPENDENT SEED 2 (FIS2).One homolog of Esc: FERTILIZATION-INDEPENDENT ENDOSPERM (FIE).Five homologs of p55: MULTICOPY SUPPRESSOR OF IRA 1 (MSI1), MSI2, MSI3, MSI4/FVE, and MSI5.

Additionally, two “readers” of H3K27me3 have been identified in *Arabidopsis*: EARLY BOLTING IN SHORT DAYS (EBS) and SHORT LIFE (SHL). Mutants of these genes exhibit similar effects to those of *PRC2* mutants, resulting in a significant reduction in H3K27me3 levels and causing a severe negative impact on plant development. Furthermore, the “eraser” mechanism of H3K27me3 has been found to utilize three specific demethylases belonging to the JmiC group, namely the Relative of Early Flowering 6 (REF6), Early FLOWERING 6 (ELF6), and JUMONJI 13 (JMJ13) [59]. Polycomb Repressive Complex 1 (PRC1) also plays a crucial role in diversifying gene regulation. One of its components, the LIKE HETEROCHROMATIN PROTEIN 1 (LHP1), is an *Arabidopsis* homolog of Heterochromatin Protein 1 (HP1) and is known for recognizing H3K27me3. This recognition indicates the cooperative relationship between PRC1 and PRC2; essentially, PRC1 “reads” H3K27me3 while PRC2 catalyzes its trimethylation. Although this mechanism is well established in animals, there is strong evidence that the plant PRC1 homolog, BAH-EMF1, performs a similar function, as demonstrated in both *Arabidopsis* and rice [60].

In plants, the trimethylation of H3K36 (H3K36me3) is predominantly found at the 5′ region of actively transcribed genes, suggesting that H3K36me3 is involved in the initiation of transcription rather than elongation. In contrast, H3K36me3 in animals is mainly found in the gene bodies of actively transcribed genes. This modification is crucial for transcription elongation and is associated with processes such as splicing and the recruitment of DNA methyltransferases, which help maintain gene body methylation. In Arabidopsis, the primary histone methyltransferase responsible for catalyzing H3K36 methylation on a global scale is EARLY FLOWERING IN SHORT DAYS (EFS)/SET DOMAIN GROUP 8 (SDG8) [61]. Additionally, SET DOMAIN GROUP 26 (SDG26) is known to specifically deposit H3K36me3 at particular gene loci [62].

Mono-, di-, and tri-methylation modifications of H3K4 on euchromatin occur through the COMPASS1 complex (Complex of Proteins Associated with SET1) and are removed by the enzyme LSD1 and JmJ domain-containing demethylases. In plants, COMPASS1 complexes include H3K4 methyltransferases along with the core subunits RBL, WDRS, and ASH2R, which have conserved functions. RBL acts as the essential scaffold for the assembly of the entire complex [51].

Overall, histone methylation mechanisms exhibit complex regulatory processes that occur at distinct target sites and different chromatin locations, influencing various aspects of transcriptome regulation. Notably, protein complexes like PRC1 and PRC2 have evolved to specifically regulate H3K27 trimethylation, a modification linked to transcriptionally inactive gene regions. Moreover, it has been demonstrated that activating marks like H3K4me3 and repressive marks such as H3K27me3 can occupy the same genomic loci. This coordination of chromatin states allows for the fine-tuning of transcription, suggesting a well-orchestrated yet specific mechanism focused on lysine residues [62]. Different enzymatic systems are employed for each lysine position and chromatin region, enabling tailored responses to various plant developmental stages and necessary defensive responses. The processes of chromatin folding and unfolding play a crucial role in determining the accessibility of transcription factors, which is essential for controlled gene expression. Together with the previously discussed epigenetic DNA methylation, this forms a sophisticated network of molecular events that finely regulate gene expression. Table 2 provides a summary of the different lysine residue methylations categorized by stage, detailing the specific enzymatic mechanisms involved and the biological roles of each methylation process.

### 3.2. Histone Arginine Methylation

In addition to lysine residue methylation, arginine residues in histones H3 and H4 also undergo methylation. This process primarily affects arginine residues at positions 2 (R2), 8 (R8), 17 (R17), and 26 (R26) in histone H3, and at position 3 (R3) in histone H4. Arginine methylation can be symmetrical or asymmetrical, depending on various guanidine configurations that are targeted. The addition of methyl groups to arginine residues is facilitated by a group of evolutionarily conserved enzymes known as arginine methyltransferases (PRMTs) [2]. While the demethylation of arginine is not fully understood, studies have identified JmjC domain enzymes as the primary erasers of methylarginines, functioning similarly to lysine demethylases [63]. Further research into the implications of arginine methylation may provide valuable insights into its biological significance, particularly regarding plant development, the cell cycle, defense mechanisms, and potentially a system that coordinates both regional and residual methylation of histones in relation to specific biological functions and gene expression.

### 3.3. Histone Acetylation

The acetylation of histones represents yet another mechanism of histone modification, which, along with other epigenetic processes, forms what is known as the “histone code” for regulating gene expression [64]. Generally, the acetylation of histone lysine residues leads to a relaxed chromatin structure, facilitating transcriptional activation, while deacetylation is associated with transcriptional repression. The dynamics of histone acetylation and deacetylation are regulated by histone acetyltransferases and histone deacetylases. These processes play a crucial role in a plant’s response to both biotic and abiotic stress, and during critical stages of plant development [50].

All four components of the octamer nucleosomes, H2A, H2B, H3, and H4, undergo a recurring process of acetylation and deacetylation at their N-terminal lysine residues. Specific lysine residues also play key roles in this regulation; for instance, H3 is modified at positions K9, K14, K23, and K27, while H4 is modified at different lysine residues, including K5, K8, K12, K16, and K20. This highlights the precise nature of the acetylation process, which is guided by acetyltransferases (HATs) and deacetylases (HDACs) [65]. HATs are complex polymeric proteins composed of several subunits, each specializing in specific histone acetylation. HATs are classified into four distinct types, with each class containing specific conserved domains that enable functions beyond histone acetylation—often referred to as their “writer function”. These additional functions can include interactions with transcription factors, zinc finger DNA-binding domains, activator proteins, and coordination with other histone-modifying enzymes [66]. The number of HATs can vary significantly between plant species. For instance, tomato plants contain 32 HATs, while Arabidopsis has only 12. Similarly, HDACs are divided into three major families that are specific to histones and are made up of several domains that help form a multiprotein repressor complex. Like HATs, the number of different types of HDACs also varies among plant species. In contrast to HATs, tomatoes have only 14 types of HDACs, whereas Arabidopsis has 18 [66,67].

### 3.4. Histone Phosphorylation

Another histone modification mechanism observed in plants involves the phosphorylation of H2A at serine 95 (H2AS95ph), which is catalyzed by MUT9-like kinase 4 (MLK4). H2AS95ph may facilitate the accumulation of the H2A variant H2A.Z, leading to the stimulation of H4 acetylation on chromatin and the activation of gene expression. This process is particularly important in processes like flowering time and plant development, as observed in Arabidopsis [68]. In addition to the established role of MLK4, recent studies have shown that MLK3 may have a similar function [69]. Moreover, it has long been known that histone H3 is also phosphorylated at serine positions 10 and 28, and threonine positions 3 and 11. For instance, during mitosis and meiosis, H3 undergoes heavy hyperphosphorylation [70].

The complexity of the “histone code” in gene regulation is reflected in the diverse histones and the specific histone residues that are modified, as summarized in Table 3. The information presented in the table clearly illustrates the specificity of the evolved histone modification mechanisms. The functional consequences of histone modifications depend on both the type of histone and the specificity of residue modifications. In addition to the extensively discussed differential reading and writing mechanisms that generate various types of modifications, these factors serve as key indicators of a well-developed and specialized molecular network responsible for the precise regulation of gene expression. This network of histone modifications is significantly influenced by the regulatory effects of microRNAs (MIRs), leading to region-specific differential DNA methylation and demethylation. Ultimately, this process determines the appropriate activation or silencing of genes, enabling plant cells to respond differently based on cell cycle-specific gene expression and to adapt to various biotic and abiotic stresses [51].

### 3.5. MIR-Dependent Regulation of Gene Expression

The vital role of non-coding RNAs in the regulation of gene expression at the post-transcriptional level has been well established in both animal and plant cells. Non-coding regulatory RNAs (ncRNAs) are primarily distinguished by their nucleotide length. Long non-coding RNAs (LNCs) are longer than 200 nucleotides, while microRNAs (miRNAs) are typically composed of 20 to 80 nucleotides in their processed, mature form [72]. In contrast to microRNAs (MIRs), all five RNA polymerases in plants can transcribe long non-coding RNAs (LNCs) [73]. However, LNCs exhibit distinct functions depending on which RNA polymerase is responsible for their transcription. For instance, LNCs transcribed by RNA polymerase IV serve as precursors for small interfering RNAs, while those transcribed by RNA polymerase V are involved in chromatin loop formation [74]. LNCs can be classified not only based on their origin of biogenesis but also according to their mechanisms of action. They can function as cis-acting or trans-acting elements that participate in DNA methylation and histone modification processes [75]. A well-characterized example of this is the LNC *COOLAIR*, an antisense transcript of the flowering locus *FLC*, which encodes a key flowering regulator. *COOLAIR* recruits the Polycomb Repressive Complex 2 (PRC2), which then represses *FLC* expression via H3K27me3 modification [75]. LNCs have been implicated in responses to both biotic and abiotic stresses, and in plant development and homeostasis. However, this area remains poorly understood, highlighting the need for further studies [76,77].

Functionally, there is an antagonistic relationship between miRNAs and LNCs: miRNAs inhibit mRNA translation, whereas LNCs can sponge up miRNAs, thereby restoring mRNA translational levels. Recent research in plants indicates a significant involvement of miRNAs in gene regulation through various pathways. This includes regulating proteins related to chromatin modifications [78]. While the primary role of miRNAs is to inhibit specific gene expression, their biological function should not always be viewed as purely inhibitory. In cases where the blocked transcript would lead to the expression of an inhibitory protein, miRNAs may actually promote the activation of gene expression [79]. Furthermore, some miRNAs have been shown to act directly as enhancer regulators and activate gene transcription, indicating their role in enhancing RNA expression and modifying histone marks [80]. However, the latter has not been identified in plants.

Plant microRNAs (miRNAs), which are a class of RNAs 21–22 nucleotides in length, play a critical role in regulating gene expression by blocking translation at the post-transcriptional level. Each miRNA specifically targets a complementary mRNA, leading to its inhibition. This mechanism is essential for plant growth and development, and for responses to environmental stress [81,82]. Plant miRNAs are transcribed by RNA polymerase II (RNA Pol II), which recognizes the 5′ cap structure (m7Gppp) and the 3′ polyadenylated regions [83]. The biogenesis of these miRNAs is a complex process that involves various processing systems, including Argonaute 1 (AGO1) and the miRNA-induced silencing complex (RISC). This process transforms an initial hairpin loop structure into a mature sequence of approximately 20–22 nucleotides that is specific to the target mRNA [84].

The preservation of microRNAs (MIRs) across various cell types, both in plants and animals, highlights their significant role in regulating the expression of specific target messenger RNAs. Additionally, chromatin modifications such as histone methylation and ubiquitination play a crucial role in controlling MIR gene expression and activity, demonstrating the collaborative interplay among these mechanisms in plants. In *Arabidopsis*, for example, the modifications at the H3K14 position have been shown to increase the accumulation and activation of specific MIRs through the action of non-repressed protein 5 (GCN5) [85]. Conversely, the Polycomb Repressive Complex 2 (PRC2), by increasing the methylation of H3K27 to the repressive H3K27me3 form, inhibits the expression of the *MIR156A/C* gene [86]. Furthermore, the protection of MIRs and prevention of their degradation has been linked to the involvement of PRL1 and MAC3 (a U-box type 3 ubiquitin ligase) [87]. In turn, as demonstrated in *Arabidopsis*, mature and functional MIRs influence the activity of various chromatin and DNA modifiers, including RNA-directed DNA methylation (RdDM) in non-coding regions (LNRs), particularly in response to pathogenic invasions [88].

## 4. Ubiquitination and Ubiquitin Ligase-Mediated Regulation

Ubiquitin (UB) is a highly conserved polypeptide composed of 76 amino acids, with slight variations across different species [89]. The biological function of ubiquitin is derived from its ability to form reversible covalent bonds with specific lysine residues on targeted proteins. This process, known as ubiquitination, can involve the attachment of a single ubiquitin molecule (monoubiquitination) or multiple ubiquitin molecules (polyubiquitination or multiubiquitination). These additional ubiquitin units can be added to any of the lysine residues found in ubiquitin’s sequence (Lys6, Lys11, Lys27, Lys29, Lys33, Lys48, Lys63) or to a methionine residue (Met1) [24]. In all eukaryotes, including plants, ubiquitination plays a role in various regulatory processes and involves interactions with other histone modification pathways, resulting in a complex “ubiquitin code” [90].

The mechanism of the monoubiquitination of histones H2B (H2Bb1) and H2A (H2Ab1) is essential and has been extensively studied in plants. This process is facilitated by specific ubiquitin–protein ligases, particularly E3-type ligases, which provide target specificity. The E3 ligases involved in this process include HISTONE MONOUBIQUITYLATION1, HISTONE MONOUBIQUITYLATION2, HUB1, and HUB2. Overall, ubiquitination in plants is a highly specific process that is regulated by the expression of specific ubiquitin genes and a large number of E3 ligase homologues. In *Arabidopsis*, 12 functional ubiquitin genes have been identified, each exhibiting biological functions related to plant development [91]. For example, UBP14 is essential for embryo development, as mutants arrest at the globular stage due to disrupted ubiquitin recycling. UBP15, on the other hand, influences leaf shape, flowering time, and seed size. Mutants exhibit smaller organs and altered morphology, while overexpression leads to larger seeds and organs [92].

The two most common types of polyubiquitination linkages observed at the cellular level are K48-linked and K63-linked polyubiquitination. K48-linked chains typically adopt a “compact” conformation with exposed I44 (Isoleucine 44) patches, directing the target proteins to the 26S proteasome for degradation. In contrast, K63-linked chains tend to adopt an “open” conformation, allowing for greater flexibility in binding associated proteins. These K63-linked chains play important non-proteolytic roles in various cellular processes, such as DNA repair and signaling pathways, mediated by kinase activation. Recent studies have also highlighted the functional diversity of the six remaining types of chain linkages, which are formed through Lys6, Lys11, Lys27, Lys29, Lys33, or Met1. These are referred to as atypical ubiquitin chains and are involved in various regulatory functions beyond proteolysis. In plants, numerous studies have identified the role of atypical ubiquitination in the immune response to pathogens [71]. The regulation of ubiquitination effects is illustrated by the interactions between H2Bub1 and the histone modifications H3K4me3 and H3K36me3. These interactions activate COMPASS-type mechanisms, which, in turn, promote gene activation. In contrast, H2Aub1 works in conjunction with H3K27me3, using PRC1 catalysis to enforce transcriptional repression and gene silencing. Thus, the ubiquitination of H2A and H2B has opposite effects, contributing to the precise regulation of homeostatic mechanisms in plant cells [93].

The ubiquitination process consists of three sequential steps. In the first step, ubiquitin activation is catalyzed by E1, the ubiquitin-activating enzyme. In the second step, the activated UB-E1 intermediate is transferred to the cysteine residue of E2, the ubiquitin-conjugating enzyme. Finally, in the third step, ubiquitin is transferred from E2 to the ε-amino group of lysine on the substrate protein. This transfer can occur directly or be facilitated indirectly by E3, the ubiquitin ligase enzyme. As a result, ubiquitin becomes attached to the substrate protein through an isopeptide linkage [94]. In Arabidopsis, there are 2 E1 enzymes, 37 E2 enzymes, and more than 1400 E3 components, which highlights the specificity conferred by the various E3 homologues [95].

Plant E3 ligases can be categorized into four groups based on their mechanisms of ubiquitin transfer: RING (Really Interesting New Gene) E3 ligases, U-box E3 ligases, HECT (Homology to E6-AP C-Terminus) E3 ligases, and RBR (RING-Between-RING) E3 ligases [96,97]. U-box E3 ligases contain a U-box domain, typically consisting of about 70 amino acids, and share structural similarities with the RING domain. However, instead of the conserved cysteine and histidine residues found in RING domains, U-box domains contain charged and polar residues. This difference stabilizes the U-box domain through a network of salt bridges and hydrogen bonds. RING and U-box E3 ligases function as platforms that bring E2 enzymes into close proximity with polypeptide substrates, which allosterically activate E2 to transfer ubiquitin to the substrate. In contrast, HECT E3 ligases are larger proteins characterized by a C-terminal HECT domain, approximately 350 amino acids in length. Within this domain, a critically conserved cysteine residue forms a thioester bond with ubiquitin transferred from E2 during the ubiquitination process, resulting in the formation of a HECT E3-ubiquitin intermediate [98]. Unlike RING E3 ligases, HECT E3 ligases directly transfer ubiquitin from E2 to the substrate via this intermediate. The RBR E3 ligases represent a hybrid of the two aforementioned types. RBR ligases contain two RING domains and interact non-covalently with the ubiquitin-E2 intermediate through their first RING domain (similar to RING E3 ligases). Subsequently, they transfer the activated ubiquitin from E2 to a conserved cysteine residue located in the second RING domain (akin to HECT E3 ligases) [99,100].

The multi-subunit group known as Cullin–RING ubiquitin ligases (CRLs) constitutes a significant portion of E3 ligases found in various organisms. CRLs are distinguished by the presence of a scaffold protein called Cullin (CUL), which recruits the RING protein (ring domain subunit RBX) at its C-terminal end and other adaptor proteins at its N-terminal end. In plants, there are three main types of CUL: CUL1, CUL3, and CUL4, as well as a CUL-like protein called APC (Anaphase-Promoting Complex) [101]. Each type of CUL binds to specific sets of adaptor molecules, which broadens the range of target proteins available for ubiquitination. Due to the diversity in substrate adaptor molecules, CRLs are further classified into four subfamilies: the SCF (S phase kinase-associated protein 1–Cullin 1–F-box), the BTB (Bric-a-brac–Tramtrack–Broad complex), the DDB (DNA Damage-Binding domain-containing), and the Cull-like protein APC (anaphase-promoting complex) [102]. Once the E3 ligase complex is assembled and the target is identified, the ubiquitin is transferred to the specific lysine residue of the targeted protein. The fate of the ubiquitin-tagged target is then determined by the interpretation of the ubiquitin code associated with that target. E3 ligases can identify substrates through specific sequence motifs, such as the D-box motif recognized by the APC/C complex and the PY motifs recognized by the KEN-box and HECT family [97]. Additionally, E3 ligases can interact with substrates through adaptor proteins. Overall, E3 ligases are vital in hormonal crosstalk and the integration of external stimuli. As will be discussed in the following chapters, ubiquitination is an important component of histone modification mechanisms and DNA methylation. Together, these mechanisms coordinate at various stages to influence plant development, immune responses, and the ability to confront biotic and abiotic stresses [103].

The process of ubiquitin conjugation to substrates is reversible and regulated by a group of enzymes known as deubiquitinating enzymes (DUBs). DUBs are crucial for cleaving peptide bonds within polyubiquitin chains, which release free ubiquitin molecules back into the cellular pool [97,98]. This recycling ensures that ubiquitin remains available for subsequent rounds of ubiquitination, helping maintain cellular ubiquitin homeostasis. In addition to recycling ubiquitin, DUBs play a regulatory role in ubiquitination events by proofreading the ubiquitination status of target proteins. They can remove ubiquitin tags from improperly tagged proteins, preventing their degradation. In plants, DUBs have been linked to various biological processes, including vacuole biogenesis, nodule formation, plant immunity, and the regulation of jasmonic acid signaling. These diverse functions underscore the importance of DUBs in plant physiology and responses to stress [104].

## 5. Histone Modifications During Seed Development and Germination

Plant development is an ongoing process that involves various stages of cell differentiation and organ formation. Key to this process is the plant’s ability to effectively regulate gene expression through precise and sophisticated regulatory mechanisms. As outlined previously, these processes rely heavily on multiple epigenetic modification mechanisms that influence the genome and chromatin structure of plant cells. However, plant genomes exhibit significantly greater diversity and complexity compared to animal genomes [105]. Most plant genomes, when compared to animal genomes, exhibit several distinct characteristics. These include increased retention of transposons, highly variable ploidy levels, a greater number of copies of transcribed genes, a higher quantity of pseudogenes and supergenes, increased nucleotide diversity, and high levels of polymorphism [104]. Additionally, plant genomes contain a significant amount of short repetitive sequences, which can take various forms such as inverted repeats, simple tandem repeats, tandem arrays, and single-copy interspersions. For instance, the genomes of sugarcane and hexaploid wheat are known to contain about 80% repetitive sequences [106,107]. The first plant genome to be sequenced was that of *Arabidopsis thaliana*, which comprises 25,489 genes and contains 14% repetitive elements within a total genome length of 125 megabases (Mbs). Recently, the smallest plant genome was identified, belonging to *Genlisea tuberosa*, with 61 Mbs and approximately 7000–8000 genes. In contrast, the plant species with the largest genome size is the Japanese canopy plant *Paris japonica*, which has an approximate genome size of 124.201 Mbs, making it 50 times larger than that of humans [108,109]. Despite the complexity of plant genomes, there are specific chromatin and DNA modification pathways that provide highly conserved mechanisms for regulating gene expression throughout the plant’s life cycle. Crucial stages of a plant’s life cycle involve seed maturation and dormancy, and seedling establishment and development. Different sets of genes are activated and suppressed at each of these stages. Research, particularly involving *Arabidopsis* and rice plants, provides strong evidence that gene activation and deactivation, along with differential gene expression, occur through a complex interplay of epigenetic modification mechanisms [110].

Histone methylation-dependent modifications play a crucial role in seed development. The trimethylation of histone H3 at lysine residue 27 (H3K27me3) by the PRC2 complex in *Arabidopsis* signifies a transition in developmental cycles and the repression of specific genes related to the endosperm. Mutations in the PRC2 complex lead to a decrease in H3K27 methylation, which results in a lower seed setting rate and various phenotypic abnormalities [111]. Interventions involving mutations in the FIS-PRC2 complex, which affects the lysine methylation mechanism, have been shown to cause developmental abnormalities in seeds of several plants, impacting various genes that are influenced downstream in the seedling pathway [13,111,112]. Furthermore, the timing of endosperm cellularization, a critical developmental step during seed maturation for both rice and *Arabidopsis*, appears to be dependent on the EMBRYONIC FLOWER 2 (EMF2) protein, which is regulated by the levels of H3K27me3 [113,114]. Seed dormancy is also influenced by the balance of various histone modifications. The interplay between H3K4me3 and H3K27me3, which have opposing roles in gene activation and deactivation, is a crucial regulatory mechanism for the state and duration of dormancy [115]. The *DELAY OF GERMINATION 1* gene (*DOG1*) is a key regulatory element of seed dormancy in *Arabidopsis* and acts as a temperature detector [115,116]. Light exposure triggers the removal of the H3K4me3 activation mark, while the repressive H3K27me3 mark accumulates on the same gene [115,117]. This highlights the well-coordinated mechanisms of gene modification control. Furthermore, Chen et al., 2020 [118] demonstrated that the PRC components CLF and LHP1 are recruited to the promoter region of *DOG1*, which accelerates the accumulation of H3K27me3 and inhibits its expression. Additionally, these components influence the activity of the REF6 demethylase, which regulates the H3K27me3 state [109]. The DOG1 pathway also represents an intriguing aspect of the dual processes of methylation and acetylation. Mutations in the H3K9 dimethylated (H3K9me2) methyltransferase *KYP/SUVH4-5* increase DOG1 activity and seed dormancy [119]. On the other hand, KYP/SUVH and the deacetylase HDA19 enhance histone H3 acetylation, also leading to increased DOG1 activity and a prolonged dormancy state [120]. Consequently, the acetylation and methylation of specific histones can exert either synergistic or antagonistic effects, depending on the particular lysine position being modified, in order to achieve the correct biological function. Seed setting is also influenced by deacetylation of histones, which is catalyzed by histone deacetylases from the HDAC family [109]. In *Arabidopsis*, it has been established that silencing the *AtHD2A* gene inhibits seed development [121]. In rice, the levels of starch in developing seeds are dependent on the activity of RICE STARCH REGULATOR 1 (RSP1). A specific NAD^+^-dependent HDAC, OsSRT1, inhibits RSP1, leading to starch accumulation [122]. Additionally, an increase in the acetylation of histones H3 and H4 was observed in maize mutants, alongside a decrease in H3K9me2, which resulted in impaired fertility [123]. Therefore, the acetylation–deacetylation process appears to play a significant, if not crucial, role in seed development. Although the mechanisms mentioned above have been observed in various plants, the existence of universal pathways for epigenetic regulation of plant development cannot be ruled out. These pathways may involve similar or corresponding plant-specific regulatory proteins that yield comparable effects from methylation and acetylation processes, ultimately leading to the coordinated regulation of plant development.

H3K4 methylation, mediated by the COMPASS1 complex, has also been shown to depend on H2B monoubiquitination (H2Bub) [124,125]. Interestingly, the knockdown of COMPASS components *SwD2* homolog from *Arabidopsis* results in multiple phenotypic changes [126], notably a reduction in the fertility rate, followed by an extended dormancy period of the seeds. These phenotypic variations are associated with decreased levels of H3K4me3 and inhibited DOG1 activity. However, this study indicates that there is no direct relationship between H2Bub1 effects on COMPASS1-mediated H3K4 methylation, as H2Bub1 and H2Bub2 function independently of H3K4 alterations in the *Arabidopsis* histone modification system [127]. This observation suggests that cross-linked histone modification systems may not always coordinate to produce a final effect. In cases of mutations and gene deficiencies, these systems may operate independently. Furthermore, in addition to the activities of “writer” systems during seed development, evidence suggests that “reader” systems also interact to create convergence. For instance, the EBS reader of H3K4me2/3 interacts with the HDAC type HDA6 to modulate gene expression. This contrasts with the independent activity of EBS towards ARXR7 and HUB proteins, which does not affect *DOG1* expression in the *Arabidopsis* genome [128]. Table 4 presents key histone modifiers and regulatory factors, highlighting their diverse roles at various stages of the plant life cycle, as studied in the reference plant Arabidopsis.

## 6. Stress-Related Epigenetic Aspects During Seed Development and Germination

Due to their stationary nature, plant growth relies on the surrounding atmospheric and soil conditions. Extreme conditions such as flooding, drought, or pathogen attacks act as stress factors that disrupt plant homeostasis. Thus, stress conditions can be classified into abiotic and biotic. Among the abiotic stress factors most influential to plant growth are salinity, water volume, temperature, light exposure, soil composition, and oxidative stress [140]. Biotic stress conditions include the interactions of plants with symbiotic and pathogenic microorganisms such as fungi, bacteria, and viruses. Plants respond to these stress factors by adjusting their homeostatic mechanisms, utilizing two complex systems of molecular pathways. These pathways are divided into (a) upstream molecular signaling, which involves factors like ABA, reactive oxygen species (ROS), ionic input (such as calcium ions), and nitrates (NO), and (b) downstream pathways that lead to adaptive modifications, including epigenetic regulation and transcription activation [141].

### 6.1. Abiotic Stress Response Mechanisms

Abiotic stress responses can affect the chromatin of plant cells on a global scale, leading to genome-wide changes in functionality. This may involve either a redistribution of these changes throughout the entire genome or localized effects on specific regions [142]. Global changes within the cellular environment include a notable increase in histone acetylation [66,143], a loss of heterochromatin chromocenter organization [144,145], and a reduction in nucleosome occupancy [146,147]. On the other hand, local changes may result in increased post-translational histone modifications, particularly methylation and demethylation, and a reduction in epigenetic marks that decrease nucleosome and DNA accessibility, e.g., to transcriptional factors and stress response genes [148] (Figure 2).

The seed germination stage of a plant’s life cycle is heavily influenced by the abiotic conditions of its immediate environment, particularly light, temperature, and water availability. These factors play a crucial role as the plant transitions from utilizing stored seed nutrients to engaging in photoautotrophy, which requires the activation of various biochemical signaling pathways and leads to different local gene expressions [149]. During germination, hormones such as gibberellic acid (GA) and abscisic acid (ABA) play significant coordinating roles. The GA pathway initiates the germination process, while the ABA pathway acts antagonistically, inhibiting germination when environmental conditions are not favorable [109,150,151]. One particularly important pathway at this stage is the Phytochrome B (PHYB)–Phytochrome-Interacting Factor 1 (PIF1) pathway. In *Arabidopsis*, PHYB destabilizes PIF1, which serves as a negative regulator of germination through the repressive factor SOMNUS (SOM). Research conducted on *Arabidopsis* has shown that the activation of the SOM gene locus inhibits germination by modulating the hormonal pathways of abscisic acid (ABA) and gibberellin (GA) [152]. PHYB enhances the expression of GA-related genes while suppressing the expression of ABA-related genes [153,154]. An epigenetic factor that influences the PHYB-PIF1 pathway in favor of PHYB involves inhibiting the ABA signal and silencing the *DOG1* gene. This process is mediated by the methylation of H3K9me2, which results from the activity of SUVH5 and HeK9 methyltransferases [137]. In contrast to the mechanism that promotes seed germination, the deacetylation of histones tends to favor PIF1 through the action of the deacetylase HDA15. This process inhibits the transcription of light-responsive germination genes [155]. Additionally, arginine methylations that affect certain genes, such as *GIBBERELLIN 3-OXIDASE 1* (*GA3ox1*) and *GA3ox2*, also suppress germination [138]. For germination to proceed, these methylations must be removed by the erasers JMJ20 and JMJ22 when PHYB is activated by light. This antagonistic relationship between PHYB, PIF1, SOM, and JMJ in seed germination creates a dual activation-inhibition mechanism that depends on the plant’s exposure to light. This highlights the importance of specific epigenetic mechanisms in an environmental pathway that is light-dependent [109]. Temperature-dependent mechanisms also highlight the crucial role of histone modifications in regulating seed germination in plants and their response to environmental signals. The EBS reader is recruited by AGL57 to the *SOM* loci, where it acts on the H3K4me3 histone modification. At elevated temperatures, the AGL57–EBS complex facilitates the acetylation of H4K5 in the *SOM* region, leading to the downregulation of germination [156]. This germination inhibitory mechanism can be reversed, allowing for the development of plant thermotolerance through the activation of HDACs and the incorporation of histone H2A.Z at the *SOM* loci [157]. Extreme drought and salt conditions can also affect seed germination through an ABA hormone-dependent mechanism. This process is regulated by the antagonistic effects of H3K9K14Ac and H3K9me2, and the activities of HDA6 and HDA19. It is generally accepted that the histone deacetylation complex 1 (HDC1) plays a critical role as the rate-limiting step in the desensitization or sensitization of germination [109,158].

### 6.2. Biotic Stress Response Mechanisms

Plants are constantly in direct contact with the surrounding microbiome, a relationship that is essential for their growth due to the intercommunication mechanisms involved. Generally, plants benefit from symbiotic microorganisms while experiencing stress from pathogenic ones (Figure 3). Plant–microbiome interactions occur in three primary compartments: (1) the rhizosphere, which is the soil surrounding the roots; (2) the endosphere, which refers to the internal contact within the plant’s organs; and (3) the phyllosphere, which represents the immediate contact between the plant and the microbiome above ground [159]. Microorganisms are attracted to plant root exudates such as carbohydrates, organic acids, phenolic compounds, coumarins, triterpenes, glucosinolates, amino acids, and other substances that help structure the plant’s surrounding microbiome [160,161]. In return, the symbiotic microbiome provides fitness advantages to the host plant and plays a significant role in helping plants cope with stress from extreme abiotic conditions, such as nutrient deprivation and drought, and biotic stress caused by pathogenic attacks [162,163,164,165].

Symbiotic microbiomes, especially plant growth-promoting bacteria (PGPB), are integral to enhancing plant health and productivity through diverse mechanisms, including epigenetic modifications. PGPB can modulate plant gene expression and the plant’s response to environmental stimuli by altering DNA methylation, histone modification, and the regulation of small RNAs [166,167]. Thus, PGPB can colonize plant roots and other tissues, establishing a mutualistic relationship that facilitates nutrient acquisition, stress tolerance, and disease resistance. For example, PGPB releases factors that induce modifications in DNA methylation patterns, thereby promoting plant growth [168]. PGPB-induced DNA methylation can also activate stress-responsive genes, thereby enhancing plants’ resilience to abiotic stresses like drought, salinity, and heavy metal toxicity [169]. Furthermore, histone modifications induced by PGPB can boost the expression of genes related to nutrient uptake and growth regulation [170]. Several plant growth-promoting bacteria (PGPB) species have emerged as pivotal agents in driving epigenetic modifications that enhance plant performance. For instance, *Pseudomonas fluorescens* has been shown to improve drought tolerance in *Arabidopsis thaliana* by altering DNA methylation patterns, resulting in the upregulation of stress-responsive genes [171]. Similarly, *Bacillus subtilis* has been reported to induce histone modifications in *Oryza sativa*, leading to enhanced growth and yield under saline conditions [172]. Another example is *Azospirillum brasilense*, which promotes root development in *Zea mays* amd thereby improving nutrient uptake and overall plant vigor [173]. Additionally, reprogramming of DNA methylation is critical for nodule development in *Medicago truncatula* [174].

The impact of biotic stress from pathogenic microorganisms has also been studied and linked to changes in the methylation patterns of plant cells. These alterations can occur in response to invasive states or due to external pathogenic inputs [175]. For instance, *Pseudomonas syringae* has been shown to induce changes in DNA methylation in *Arabidopsis thaliana*. These changes are linked to the activation of defense-related genes, particularly those involved in the salicylic acid (SA) pathway, which is crucial for plant immunity [46]. The fungal pathogen *Verticillium dahliae* is well known for causing vascular wilt in several plants, including cotton. It induces changes in DNA methylation patterns, which are believed to enhance the plant’s defense mechanisms by modulating the expression of genes responsible for pathogen recognition and response [176]. Similarly, viral infections, such as those caused by the tomato yellow leaf curl virus (TYLCV), also lead to modifications in DNA methylation in tomato plants. These changes are associated with the silencing of viral genes and the activation of host defense pathways, underscoring the importance of epigenetic regulation in antiviral responses [177]. Additionally, the oomycete pathogen *Phytophthora infestans*, which causes late blight in potatoes, triggers DNA methylation changes that regulate genes involved in the jasmonic acid (JA) and ethylene (ET) signaling pathways, both of which are critical for plant defense against such infections [178]. Furthermore, root-knot nematodes (*Meloidogyne* spp.) have been shown to alter DNA methylation patterns in host plants like tomatoes, impacting the regulation of genes related to cell wall modification and defense responses that are essential in restricting nematode invasion and reproduction [179]. Overall, these findings highlight the crucial role of DNA methylation as a regulatory mechanism in plant–pathogen interactions across various pathogen types.

Evidence suggests that DNA methylation, in conjunction with specific histone modifications, plays a crucial role in coordinating the plant cell’s immune response to biotic stress. Specifically, there is a direct association between DNA methylation on transposable elements (TEs) and the histone modification H3K9me2, which leads to TE silencing [180]. Research, particularly on *Arabidopsis*, in which extensive studies have been conducted, has shown that the dynamics of DNA methylation and demethylation depend on RNA-directed DNA methylation (RdDM) and the contribution of heterochromatic small interfering RNAs (siRNAs) derived from transposable elements (TEs). This is important because transcriptional activating factors cannot bind to methylated DNA, resulting in the suppression of specific gene expression [88]. *Arabidopsis* mutants with high levels of hypomethylation exhibit greater resistance to pathogenic infections [181], while mutants lacking the H3K36 methyltransferase SDG8 are more susceptible to infection [182]. Also, *Arabidopsis* mutants with impaired DNA methylation demonstrated significantly lower susceptibility to *Hyaloperonospora arabidopsidis* (*Hpa*) infections [183]. In contrast, mutants lacking the ATXR7 methyltransferase were more susceptible to the same disease due to their inability to methylate H3K4me3 [184]. Furthermore, studies with other mutants indicate that both DNA methylation and H3K9me2 are involved in regulating resistance to *Pseudomonas syringae* (*Pst*) infection [185], and that the INCREASE IN BONSAI METHYLATION 1 (IBM1), which acts on H3K9 by removing both mono- and dimethylation units, displays high activity and enhances defense responses against *Pst* infection in Arabidopsis [186].

H2Bub1 monoubiquitination has also been shown to enhance plant immune mechanisms by regulating the activity of SNC1 (SUPPRESSOR OF NPR1-1 CONSTITUTIVE 1), which subsequently boosts immune responses to infections. In Bonsai, the E3 ubiquitin ligases HUB1 and HUB2 are essential for the expression of *SNC1*, a disease resistance gene critical for plant immunity. Mutants lacking *HUB1* or *HUB2* exhibit significantly reduced *SNC1* expression and suppressed constitutive immune responses during infections [187]. This reduction is linked to their inability to mediate monoubiquitination of histone H2B (H2Bub1) at the *SNC1* locus, a modification necessary for its transcriptional activation. Additionally, H3K4 and H3K36 methylation have been directly linked to H2Bub1 and SNC1 activity, suggesting crosstalk between these two types of histone modification [188].

Thus, the state of genomic and histone methylation is a crucial factor in a plant’s defense responses. Similar to abiotic stress, various histone modifications, DNA methylation, and MIR activity seem to regulate the plant’s genetic potential to adapt and respond to infections through different pathways. Interestingly, these epigenetic changes induced by biotic stress may sometimes be transmitted to the progeny as transgenerational priming, underscoring the importance of chromatin modifications in regulating immune responses in plants [181,183,188].

## 7. Conclusions

DNA methylation, histone modifications, ubiquitination, and miRNA interference are key epigenetic regulators in plants. These mechanisms interact to control gene expression and enable stress memory and transgenerational adaptation, ultimately helping plants to develop and acclimate to environmental stresses. DNA methylation and histone modifications influence each other in a feedback loop, while miRNAs and ubiquitination regulate related pathways post-transcriptionally.

However, significant challenges persist in understanding the context-dependent effects, the crosstalk between these mechanisms, and the long-term stability of epigenetic modifications across various plant species and environments. Some key knowledge gaps include understanding how specific epigenetic marks are directed to specific loci during development or in response to stress, the mechanisms that integrate various epigenetic pathways, the stability of stress-induced epigenetic changes across generations, and how epigenetic regulators differentiate between cues that signal developmental processes and those that indicate stress responses. Last but not least, scientists are also troubled by the unresolved questions of whether these epigenetic edits can be tissue-specific or environmentally stable, and how polyploid genomes coordinate epigenetic regulation.

As research continues to unravel the molecular intricacies of epigenetic regulation, it becomes increasingly clear that these processes also hold immense potential for applications in crop improvement and sustainable agriculture. Epigenetic editing technologies, such as CRISPR-based tools, offer promising strategies for precisely manipulating gene expression. Future research could prioritize integrating multi-omics approaches to explore the interactions among epigenetic modulators and their functional outcomes, single-cell epigenomics to map sapatiotemporal epigenetic changes, field studies to assess epigenetic memory under real-world stress conditions, and evolutionary epigenetics to compare mechanisms across plant species. Additionally, creating high-resolution epigenomic maps and advanced computational models could be essential for predicting the phenotypic consequences of specific epigenetic changes and identifying key regulatory hubs. Collaboration among diverse scientific fields is essential for maximizing the potential of epigenetic research and transforming these insights into effective applications. By leveraging these epigenetic regulatory capabilities, researchers could revolutionize plant breeding by enabling the development of crops with enhanced stress tolerance, improved yield, and optimized nutritional profiles, addressing the pressing challenges posed by climate change and food security.

## Figures and Tables

**Figure 1 ijms-26-04700-f001:**
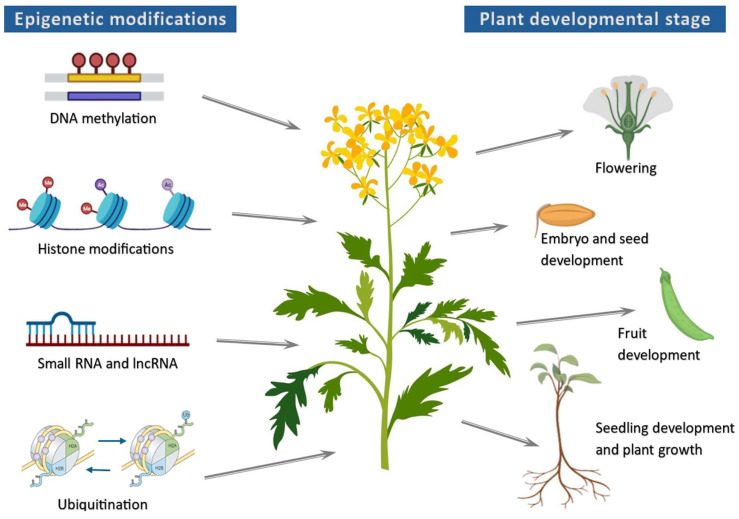
Importance of epigenetic processes to key plant processes. The illustration highlights the significance of epigenetic mechanisms in the regulation of crucial agronomic traits in crops. Different plant processes, such as germination, growth, flowering, and fruit development, are impacted by DNA methylation, histone modifications, and small RNAs in gene expression, which subsequently influence crop productivity, yields, and quality.

**Figure 2 ijms-26-04700-f002:**
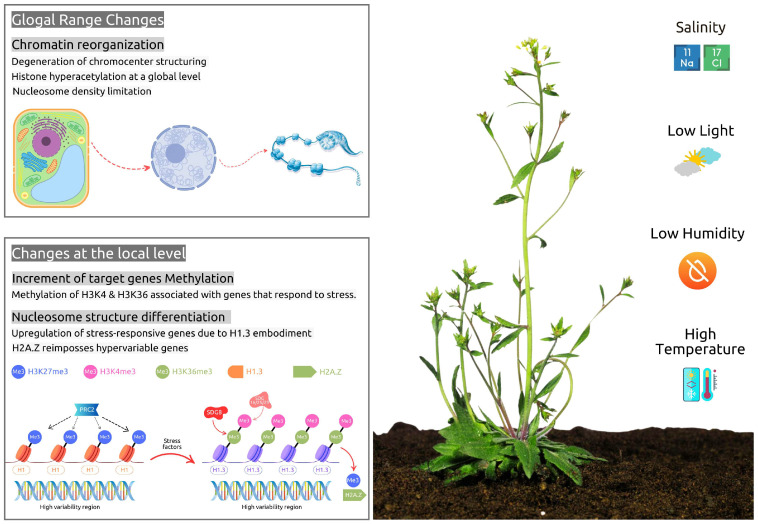
Epigenetic changes under abiotic stress. In response to abiotic stress conditions, global epigenetic changes may occur, such as chromatin reorganization. Changes at the local level involve methylation of specific target gene loci and changes in nucleosome composition.

**Figure 3 ijms-26-04700-f003:**
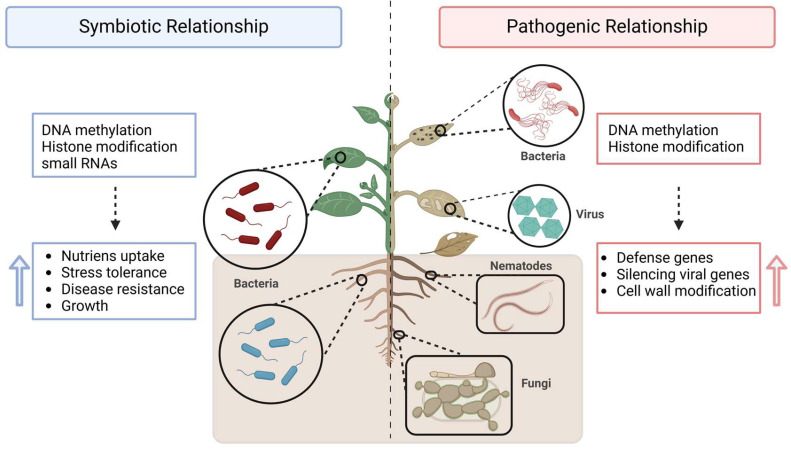
Epigenetic modifications under biotic stress. The interaction between plants and microorganisms occurs in different plant organs, such as roots and leaves, and can be either symbiotic or pathogenic. Symbiotic interactions (**left side**) initiate a cascade of epigenetic modifications that benefit the plant by enhancing nutrient uptake, stress tolerance, disease resistance, and growth (upward blue arrow). Conversely, pathogenic interactions (**right side**) also trigger epigenetic modifications as part of the plant’s defense response, leading to the activation of defense-related genes, silencing of viral genes, and remodeling of the cell wall to mitigate infection (upward red arrow). Figure created with www.BioRender.com (accessed on 21 April 2025).

**Table 1 ijms-26-04700-t001:** Targeted genomic regions for methylation and the specific functions arising from this epigenetic effect.

Targeted Genomic Region	Specific Function	References
Gene regulatory regions (promoters and enhancers)	Transcriptional gene silencing (TGS)	[35]
Intronic TEs and DNA repeats	Alternative mRNA splicing and alternative mRNA polyadenylation	[13]
Euchromatin in regions read by SUVH1 and SUVH3 methyl-readers	Proximal gene expression activation of ROS1 and DNAJ1 and 2	[13,15,39]
In GC-rich areas of gene exonic regions	Suppresses intragenic antisense transcripts	[48]

**Table 2 ijms-26-04700-t002:** Synopsis of major characteristics of methylation/demethylation mechanisms of lysine residues of histone 3 in plants with reference to studies performed on Arabidopsis. For more information, please refer to the text.

Methylation	Readers	Writers	Erasers	Biological Role
H3K9me1 and 2	ADCP1/AGDP1	KRYPTONITE (KYP)/SUVH4, SUVH5, and SUVH6	JmiC	Repressive state of heterochromatin, transposon silencing
H3K27me1		ARABIDOPSIS TRITHORAX-RELATED PROTEIN 5 (ATXR5) and ATXR6	JmiC	Heterochromatin active state
H3K27me3	PRC1, EBS, and SHL	PRC2	JmiC group (REF6, ELF6, JMJ13)	Repressive state of euchromatin
H3K4me1/2/3		COMPASS-1	LSD1, JmiC group	Euchromatin activation
H3K36me3		SDG8 and SDG26		Gene activation

**Table 3 ijms-26-04700-t003:** List of histones and the specific residues modified, resulting in different types of modifications.

Histone	Residue	Type of Modification	References
H3	LYS9 (K9)	Me1 and me2	[51]
H3	LYS27	Me1 and me3	[56,59,62]
H3	LYS4	Me1,2,3	[51,62]
H3	LYS36	Me3	[62]
H3	LYS9,14,23,27	Acetylation	[50,65]
H3	SER10 and SER28, THR3 and THR11	Phosphorylation	[70]
H4	LYS5,8,12,16	Acetylation	[65]
H2A	SER95	Phosphorylation	[68]
H2A	LYS48,63 and MET1	Monoubiquitination	[24]
H2A	LYS48,63	Polyubiquitination	[24]
H2B	LYS6,11,27,29,33	Monoubiquitination	[71]

**Table 4 ijms-26-04700-t004:** Key histone modifiers and regulatory factors, highlighting their diverse roles at various stages of the plant life cycle, as studied in the reference plant Arabidopsis.

Gene	Activity	Functional Outcome	References
HD2A	Deacetylase	Regulates seed development and germination	[129]
SD2C	Deacetylase	Regulates seed germination	[130]
HDA6	Deacetylase	Regulates seed germination and establishment	[130,131]
EMF 1/2	Embryonic flowering reduces H3K27me3 methylation	Seed enlargement	[109,132]
		Repression of maturation following germination	[13,133]
DOG1	Delay of dormancy gene 1	Temperature detector for dormancy state	[115,134]
EFS	Early flowering in short days, H3K36 writer	Regulator of seed germination and seedling establishment	[135]
CLF	Curly leaf gene, PRC2 component, H3K27me3 writer	Involved in regulation of seed dormancy, development, and seedling establishment	[136]
SUVH 4,5	HeK9me2 writers	Regulates seed dormancy and seed germination	[137]
ATX1	Arabidopsis trithorax related H3K4 reader	Counteracts CLF repression during seedling establishment	[109]
JMJ family	Histone demethylases	Enhances ABA response, positive regulation of seed germination	[109,138]
REF6	Relative of early flowering 6, H3K27me3 demethylases	Decreases rate of seed dormancy	[139]
HUB1,2	Histone monoubiquitination	Activates DOG1 and seed dormancy	[109]
FIS2	Fertilization-independent seed complexes with PRC2	Enhance seed development	[133]
EBS	Early bolting in short days, H3K4me2/3 writer, interacts with HDAC	Increases seed dormancy and decreases seed generation	[109,128]

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
