# Peer review of "A Brief Overview of the Epigenetic Regulatory Mechanisms in Plants"

_ijms, 2025, doi:10.3390/ijms26104700_

Round 1

Reviewer 1 Report

Comments and Suggestions for Authors

MS-No. 3488082-peer-review-v1-25-2-21

The review article under the title A Brief Overview of the Epigenetic Regulatory Mechanisms in Plants is well described and follow on the scope with the IJMS journal. However, there have minor and major concern which the author must be improve before the publication. Following of my comments allow for the clarification of sentences and incorporation of positive explanations for each of the comments provided.

General Comments

Overall the review described well and positive way, however, in the abstract I don’t find any informative description about the epigenetic mechanism which play crucial role for plant development and productivity. The author must be added some information like this way. Additionally, also add some mechanistic statement in the text with relative example to easily understand the scope of epigenetic mechanism to enhance plant development and resist the abiotic stress.  

Specific Comments

In abstract line 15-17 should be simplify it for clarification.

Revise all the keywords according to the aim of the study.

Why epigenetic modification is important to plant development and tolerance to abiotic stress?

Add more comprehensive and mechanistic examples that what mechanism needed to increase the plant productivity.

Provide the recent gaps and advancements to explain in the aims of this review.

Author must be disclose the critical analysis of different published articles.

Line 135-147 add relevant references in the text.

The important part of the epigenetic mechanism like lncRNA is missing.

What are the key unresolved questions that future research should focus on?

The author must be indicated the key finding gaps and future direction in the conclusion section rather than repetition of the discussion and general statements. It should clearly summarize major findings, knowledge gaps, and future directions.

The table of the review is not acceptable. The author must be clearly take a good example from the published papers their role, species, gene, and citation etc. Revise all the tables with informative potential.

I do not see any figures in the review. Without figures, the review is incomplete. The author must add 2–4 different mechanistic figures to help readers understand the content more easily.  

Author Response

Response to Reviewer #1 for Manuscript ID: ijms-3488082 (Tresas et. al.)

Summary

We thank the Reviewer for the excellent review and his/her constructive criticism. The Reviewer brought up several points in the MS, which we agree with. The Reviewer helped us to correct several mistakes and oversights and to significantly improve the manuscript. We have addressed all comments and hope that the Reviewer is satisfied with the performed changes and additions. Detailed responses and corresponding revisions are highlighted in green in the re-submitted files, and summarized here as follows.

General Comments

Comment: [Overall the review described well and positive way, however, in the abstract I don’t find any informative description about the epigenetic mechanism which play crucial role for plant development and productivity. The author must be added some information like this way.]

Response: Thank you for pointing this out. We have revised the entire Abstract, which now includes the information requested by the reviewer.

Comment: [Additionally, also add some mechanistic statement in the text with relative example to easily understand the scope of epigenetic mechanism to enhance plant development and resist the abiotic stress.]

Response: The text includes various mechanistic statements and examples that illustrate how the different epigenetic mechanisms contribute to plant development and stress resistance. Please allow us to state that a more detailed description of the molecular function of each gene/complex and a critical analysis of all the references listed is beyond the scope of this review (“A Brief Overview of the Epigenetic…”) and may confuse the reader.

Specific Comments

Comment: [In abstract line 15-17 should be simplify it for clarification.]

Response: As requested, we have clarified this section of the “Abstract” for better understanding.

Comment: [Revise all the keywords according to the aim of the study.]

Response: We have revised the list of keywords.

Comment: [Add more comprehensive and mechanistic examples that what mechanism needed to increase the plant productivity.]

Response: In this review, we define plant productivity as a broad concept that encompasses factors such as plant growth, flowering, seed set, and fruit development. These aspects are influenced by various epigenetic mechanisms that regulate gene expression in plants, along with their responses to environmental stimuli, nutrient acquisition, stress tolerance, and disease resistance. As a result, a plant's ability to better adapt to its environment typically leads to increased productivity.

Comment: [Provide the recent gaps and advancements to explain in the aims of this review.]

Response: We agree that this information was missing, and thank the Reviewer for the comment. We have elaborated the last paragraph of the “Introduction” (lines 134-141) to include this information requested by the Reviewer.

Comment: [Line 135-147 add relevant references in the text.]

Response: We thank the Reviewer for pointing out the lack of references in this section. We have included the relevant references in this section. Moreover, and in response to the Reviewers’ feedback, we thoroughly revised the entire manuscript and meticulously incorporated additional bibliography into various parts of the MS, ensuring a more comprehensive presentation of our work.

Comment: [The important part of the epigenetic mechanism like lncRNA is missing.]

Response: We have elaborated section 3.5 and have included the relevant information concerning the lncRNAs (lines 406-420) as requested by the Reviewer.

Comment: [The author must be indicated the key finding gaps and future direction in the conclusion section rather than repetition of the discussion and general statements. It should clearly summarize major findings, knowledge gaps, and future directions……What are the key unresolved questions that future research should focus on?]

Response: We fully agree with the Reviewer that the "Conclusion" section did not fulfill its intended purpose. As a result, we have completely revised the "Conclusions" to clearly summarize the key findings, identify knowledge gaps, and outline future research directions, as requested by the Reviewer.

Comment: [The table of the review is not acceptable. The author must be clearly take a good example from the published papers their role, species, gene, and citation etc. Revise all the tables with informative potential.]

Response: We appreciate the Reviewer’s comment and have made comprehensive revisions to all tables to ensure they align with the feedback provided. Furthermore, we have added an additional Table (Table 1) in Section 2 (DNA Methylation).

Comment: [I do not see any figures in the review. Without figures, the review is incomplete. The author must add 2–4 different mechanistic figures to help readers understand the content more easily.]

Response: We acknowledge the Reviewer’s comment and have included three figures in the revised manuscript to enhance readers' understanding, as per the Reviewer's request.

We believe that we have effectively addressed all comments and hope that the Reviewer will find the revised version of our manuscript satisfactory and improved.

Reviewer 2 Report

Comments and Suggestions for Authors

Dear Editor and Authors,

I appreciate the opportunity to review the manuscript titled "Brief Overview of the Epigenetic Regulatory Mechanisms in Plants" submitted to the International Journal of Molecular Sciences This manuscript presents an extensive and valuable review of epigenetic regulatory mechanisms in plants, covering a broad spectrum of topics, including DNA methylation, histone modifications, and non-coding RNA-mediated gene regulation. The authors have compiled a vast amount of information, demonstrated a comprehensive understanding of the subject, and provided a resource that will be highly beneficial to researchers in the field of plant epigenetics.

However, while this review is well-researched and contains a valuable compilation of data, there are some areas where improvements could enhance readability, clarity, and overall impact. For instance:

  • the manuscript contains many long and complex sentences that could be simplified for better readability.
  • Several paragraphs contain detailed explanations but lack direct citations to support the claims. Every key statement, particularly those discussing experimental evidence or major conclusions from the literature, should be backed by appropriate references.
  • In some points, like lines 426-428, there are comparisons between other groups of organisms like yeast and humans; perhaps this could be rephrased to focus directly on the mechanism of plants, avoiding confusion.
  • The manuscript would benefit greatly from additional visual elements like conceptual schemes -illustrating key pathways and mechanisms that would make complex interactions easier to understand; more tables summarizing information and possible figures as high-quality diagrams representing the regulatory pathways described in the review.

Overall, this review is a highly informative and well-researched contribution to the field of plant epigenetics. With improvements in readability, citation completeness, and structural organization, it has the potential to serve as an excellent reference for researchers. I encourage the authors to consider simplifying certain sections, adding more visual elements, verifying citations, and ensuring that all information is clearly referenced. So, I would recommend a minor review for acceptance of the manuscript, and I also want to express my appreciation for the efforts of the authors in compiling such an extensive review.

Author Response

Response to Reviewer #2 for Manuscript ID: ijms-3488082 (Tresas et. al.)

Summary

We thank the Reviewer for the excellent review and his/her constructive criticism. The Reviewer brought up several points in the MS, which we agree with. The Reviewer helped us to correct several mistakes and oversights and to significantly improve the manuscript. We have addressed all comments and hope that the Reviewer is satisfied with the performed changes and additions. Detailed responses and corresponding revisions are highlighted in green in the re-submitted files, and summarized here as follows.

Comment: the manuscript contains many long and complex sentences that could be simplified for better readability.]

Response: We very much appreciate the Reviewer for the valuable suggestions and corrections included in the attached PDF file. Their input was crucial in enhancing the clarity of several parts of the manuscript. We have implemented all the suggestions and corrections, which are now reflected in the revised version of our manuscript.

Comment: [Several paragraphs contain detailed explanations but lack direct citations to support the claims. Every key statement, particularly those discussing experimental evidence or major conclusions from the literature, should be backed by appropriate references.]

Response: We acknowledge the Reviewer’s comment and are thankful for pointing out the lack of references in some parts of the manuscript. In response to the Reviewers’ feedback, we thoroughly revised the entire manuscript and meticulously incorporated additional bibliography into various parts of the MS, ensuring that all data provided is backed up by appropriate references.

Comment: [In lines 426-428, there are comparisons between other groups of organisms like yeast and humans; perhaps this could be rephrased to focus directly on the mechanism of plants, avoiding confusion.]

Response: We think that the Reviewer is referring to lines 439-441 (480-482 in the revised MS) instead of lines 426-428. These are the only lines that mention differences between other groups of organisms, like yeast and humans. Nevertheless, we agree with the Reviewer that the statement mentioning the differences between yeast and human is irrelevant. We have therefore deleted the sentence “The difference, for instance, between yeast and human ubiquitin lies in just three amino acids.

Comment: [The manuscript would benefit greatly from additional visual elements like conceptual schemes -illustrating key pathways and mechanisms that would make complex interactions easier to understand; more tables summarizing information and possible figures as high-quality diagrams representing the regulatory pathways described in the review.]

Response: We appreciate the Reviewer’s comment and have made comprehensive revisions to all tables to ensure they align with the feedback provided. Furthermore, we have added an additional Table (Table 1) in Section 2 (DNA Methylation). We have also included three figures in the revised manuscript to enhance readers' understanding, as per the Reviewer's request.

We believe that we have effectively addressed all comments and hope that the Reviewer will find the revised version of our manuscript satisfactory and improved.

Round 2

Reviewer 1 Report

Comments and Suggestions for Authors

I have reviewed the revise version review article which show significant improvement than the first version and the author addressed all of my concern positively. Now I am pleased to confident that the revise version manuscript is suitable for publication in IJMS. 

Thank you